# Gender Differences in the Attitudes of Parents Living with Adult Children with Schizophrenia

**DOI:** 10.3390/healthcare9070836

**Published:** 2021-07-01

**Authors:** Tzu-Pei Yeh, Ying-Wen Lin, Hsing-Yu Liu, Tzu-Ching Yang, Wen-Jiuan Yen, Wei-Fen Ma

**Affiliations:** 1School of Nursing, China Medical University, Taichung 406040, Taiwan; tzupeiyeh@mail.cmu.edu.tw; 2Nursing Department, China Medical University Hospital, Taichung, 404332, Taiwan; 3Department of Nursing, National Cheng Kung University Hospital Dou-Liou Branch, Yunlin 701401, Taiwan; apple200261@gmail.com; 4Department of Nursing, Tsaotun Psychiatric Center, Ministry of Health and Welfare, Nantou 54249, Taiwan; hsingyuliu76@gmail.com; 5Department of Nursing, Changhua Christian Hospital, Changhua 53336, Taiwan; gean.yang@msa.hinet.net; 6Department of Nursing, Chung-Shan Medical University, Taichung 40201, Taiwan; wyen@csmu.edu.tw; 7Program for Health Science and Industry, China Medical University, Taichung 406404, Taiwan

**Keywords:** schizophrenia, social roles, gender difference, parent, primary caregiver

## Abstract

Background: This paper explored the gender differences in the attitudes of parents toward taking care of their adult daughters or sons with schizophrenia, and focused on how parents define and think about the roles of their children, and how they cope with fulfilling the expected roles. Methods: Qualitative research design and purpose sampling were used to enroll parents who had adult patients with schizophrenia at a medical center in central Taiwan. Semi-structured in-depth interviews were conducted and content analysis was used to analyze the data. Results: Seven main themes emerged from the data provided by ten parents living with adult children with schizophrenia. Three themes that focused on gender difference are listed: parents continue to expect their sons with schizophrenia to carry on the family name; society as a whole expects males to be the “head of the family”; male family members are supposed to assume the responsibility of caring for siblings with schizophrenia. Conclusions: The results of the study could help clinical professionals to understand and have greater empathy with regard to the difficulties for families and the concerns of parents taking care of their children with schizophrenia in the specific context of Chinese culture, and to provide more efficient and responsive assistance.

## 1. Introduction

Living in social groups, most people are assigned respective social roles inevitably requiring them to behave in ways that meet the expectations of those roles [1]. However, people with mental health conditions are seldom able to fulfill their expected social roles effectively due to physical and cognitive decline caused by illness [2,3]. They are accordingly dependent on their family members as their primary caregivers [4,5]. As indicated by previous studies, living with mental illness does not prevent people from fulfilling their social, roles shaped by their cultures including gender roles [6,7]. Nevertheless, how cultural, social, and/or gender factors affect the attitudes and expectations from family members as primary caregivers toward their loved ones living with mental illness remains unclear. Clinical professionals can therefore become reliable supporters when family caregivers need help in coping with the unmet expectations of their ill family members, improving family relationships, and enhancing quality of life.

From the very moment of their birth, individuals start to develop male or female temperaments under the influence of their gender during the process of socialization. They learn to play different roles in line with their gender, as prescribed by cultural expectations [8]. An individual, therefore, tends to behave in expected ways in society and dictated by his or her social roles [9]. Both society and culture assign expectations and responsibilities to individuals based on their gender [10], hence the belief that fulfilling one’s social roles, such as honoring the responsibilities of human reproduction and parental responsibility through hard work, makes one’s life more meaningful [11,12].

According to the 2018 Ministry of Health and Welfare (MHW) statistics, there were 1,173,978 disability card holders in Taiwan; 127,591 (10.87%) were patients with chronic mental disorders, and those aged between 30–59 years accounted for approximately 68.65% of these [13]. People in this age have certain social roles including as a wage earner, spouse, or parent. However, due to the impacts of their mental illness such as cognitive deficit and functional impairment, they are unlikely to be able to continue fulfilling their social roles after returning to the community. This inability can trigger drastic changes in their relationships with family members, disrupting familial harmony that relies to a great extent on every member’s successful performance of his or her expected roles [14,15]. In western countries, arranged marriages are quite rare, even in people with mental illness demanding a certain degree of social skills in seeking partners [16].

In addition to accepting the fact that they are living with a patient with a mental illness, family members need to assume the responsibility of taking care of the patient and cope with the stigma attached by society to mental illness. Both can cause family members tremendous pressure and affect their health; in extreme cases, they may be overstressed that they are unable to continue taking care of their loved ones with a mental illness [17,18,19]. People living with mental illness are often treated unfairly and in an unfriendly way by family and society, or in the workplace, shunned as if they have the plague by most people. As a result, they can barely build close relationships with others. However, approximately more than 50% of caregivers hope that their family members with mental illness will get married, lessening the stigma from outsiders; most importantly, they hope the spouse will take care of the family member with a mental illness when they are not able to offer care anymore [20].

It is difficult for people in Chinese-speaking countries to lead their lives in defiance of the traditional roles and expectations prescribed by their culture, including the obligation of carrying on the family name. Chinese culture places great emphasis on gender roles. Gender role expectations affect nearly every aspect of an individual’s life, including attitudes, perceptions, responsibilities, and rights in a wide setting from family to school and the workplace [21,22]. Although mental illness patients of different genders seem to have very close experiences of stigma in western countries, female patients are still considered to be less dangerous and their emotional and passive behaviors are more acceptable [23]. People with mental illnesses find it difficult to fulfill their expected roles. Therefore, for parents as the primary caregivers of adult children with schizophrenia, taking care of their sons and daughters inevitably involves the need to cope with the aforementioned difficulty/inability, a crucial issue that has remained under-studied.

## 2. Materials and Methods

This study adopted a qualitative descriptive design [24] to examine what expectations parents taking care of adult children with schizophrenia have of their sons and daughters’ roles and how they cope with the difficulties their children have in fulfilling the expected roles prescribed by traditional Chinese culture. In-depth interviews were conducted to investigate and interpret their life contexts and experiences as well as impacts [25,26].

### 2.1. Research Subjects

Purposive sampling [24] was used to enroll parents who were primary caregivers of their adult sons or daughters with schizophrenia who were receiving outpatient or home care in a medical center in central Taiwan. Enrolled subjects needed to be able to communicate in both Chinese and Taiwanese, living with their children with schizophrenia and taking care of them for at least one year, and agree to sign informed consents.

### 2.2. Data Collection

Data was collected through face-to-face in-depth interviews. Interviews were recorded and supplemented by on-site note-taking of non-verbal cues. A complete interview lasted approximately one hour and was conducted at a venue familiar to the subject or capable of making the subject feel relaxed and comfortable for talking about his or her experiences and feelings. Please check Table 1 for the interview guide.

### 2.3. Data Analysis

Recorded interviews were transcribed verbatim into written texts, and the transcription was performed immediately after data collection to reduce memory- and subjectivity-related potential errors. Every written text of the interview was reviewed for editing a reflective journal. Content analysis [27] was then conducted with the following six steps [24]: (1) listen to each recorded interview repeatedly to transcribe it verbatim into a written text, (2) read each transcript repeatedly and take note of relevant and meaningful passages for completing open coding, (3) reviewing the relevant and meaningful passages to identify sentences capable of representing the interviewee’s experiences and feelings, (4) categorize representative sentences to develop common themes, name the categories or themes, and identify sub-themes, (5) re-examine the data to verify if the common themes are categorized and named as they should be, make necessary revisions, and infer meaningful contexts, and (6) describe them in a comprehensive manner to present study results based on the categorized themes.

### 2.4. Study Rigor

The study strived to ensure the reliability of qualitative research by following the four criteria—credibility, transferability, dependability, and confirmability—proposed by [28] to refine the concept of trustworthiness. The researcher, equipped with five years experience in clinical mental health nursing, studied qualitative research prior to conducting the study. Interviews were recorded and transcribed by the researcher into written texts, which were reviewed carefully and comprehensively to facilitate the writing of reflective journals to avoid the researcher’s personal experiences and attitudes from affecting data interpretation and analysis. Analyzed results were examined and confirmed by another experienced researcher to ensure their accuracy and completeness. Interview guidelines and open questions were applied to encourage participants to describe their experiences and feelings. The researcher discussed the analyzed results with the research team members during the entire process of data analysis to keep subjective values and preconceived notions from interfering and misleading data interpretation, thereby enhancing the confirmability of study results.

### 2.5. Ethical Considerations

The institutional review board of the host medical center approved the study (Document No: CMUH105-REC1-106). Prior to conducting interviews, the objective of the study and the procedures of the interview were explained clearly to the enrolled subjects to obtain their informed consents. The subject names were transferred into anonymity in the report for confidentiality. All recordings and transcribed interview data were kept in an encrypted digital device and used only for academic and research purposes.

## 3. Results

### 3.1. Demographic Information of Research Subjects

Ten parents as primary caregivers of their adult children with schizophrenia participated in this study. Three of them were fathers and seven were mothers; they were aged between 53–90 years old. Their adult children with schizophrenia included eight males and two females. Five enrolled subjects graduated from junior high school. The duration of taking care of their adult children ranged from three to twenty-five years with five subjects playing the role of primary caregiver for as long as twenty years. Please check Table 2 for the essential demographic information of the research subjects.

### 3.2. Seven Main Themes

The study examines the influences of traditional Chinese culture on the attitudes of parents living with adult sons and daughters with schizophrenia and identifies 7 main themes based on the analysis of the data provided by 10 parents (Table 3). Four of these themes showed no gender difference in parents’ attitudes toward their children, while gender difference influences were observed in the remaining three themes (Figure 1). Gender as a label in Chinese culture appears to affect not only patients with schizophrenia themselves but also the attitudes of their parents as primary caregivers.

The four themes marked with no gender difference in parents’ attitudes toward their ill children are summarized as follows:

#### 3.2.1. Theme One: Parents Do Recognize the Need of Re-Examining the Roles and Wishes Which Their Children Are Unlikely to Be Able to Fulfill

Most parents care about their children and expect them to be able to complete the “missions” during different stages of life. Due to their child’s illness, however, many of the parents’ expectations become unrealistic. Parents are forced to reduce or even relinquish their expectations of their ill sons and daughters, leaving everlasting regrets in their lives. Three sub-themes were included:

##### Experiences of Shame and Learn to Accept

Through the interviews, parents identified their adult children with schizophrenia as irreplaceable family members. Parents do experience shame and feel sorry for the stigma imposed on their children. They learn to accept their imperfections.


*“It’s nothing to speak out, and we try our best not to bring it up…Some people, you know, would warn others against getting too close to us…they make sure to avoid contact with us in fear of being affected”.*
(Subject J, a mother taking care of a son with schizophrenia)


*“There lives a nutcase, a sick man,” people would say. We on our part are afraid of telling people that he is mentally ill. But it doesn’t really matter, as long as he gets better, right? He is not a thief or robber. He is just mentally ill…”*
(Subject I, a father taking care of a son with schizophrenia)

##### Forced to Give Up Expectations

Parents feel forced to give up expectations of their children with schizophrenia. They tend to believe that, because of schizophrenia, their children have no hope for marriage.


*“You just have the feeling that she is…there is no hope left for her! There is nothing you can expect from her, nothing…”*
(Subject D, a mother taking care of a daughter with schizophrenia)


*“With this illness, I wouldn’t dare to find him a wife. His child will be unhealthy like him. That’s a terrible thing to everyone, almost a sin. It’s not right.”*
(Subject G, a mother taking care of a son with schizophrenia)

##### Fail to Be Independent by Their Own

Parents’ expect that their children may develop skills for independence, but often fail to obtain supporting opinions from medical professionals.


*“I’ve kept trying to help her to learn a practical skill so she can…you know…but the doctor says that we should focus on keeping her stable without deterioration.”*
(Subject A, a father taking care of a daughter with schizophrenia)

#### 3.2.2. Theme Two: Hospitalization Gives Parents a Break after Long-Term Mental and Physical Burnout

Family members, especially parents as primary caregivers, are understandably upset and desperate toward their loved ones with schizophrenia, as they have to live in constant fear of a continuous or relapsing status of psychosis that can trigger overwhelming emotion responses. Stress, exhaustion, and a sense of helplessness keep accumulating, resulting in almost unbearable long-term burnout. Hospitalization thus becomes a paradise-like exit promising temporary breaks. Four sub-themes were included:

##### Parents Feel Emotionally and Physically Worn Out

Parents, overwhelmed by recurrent psychotic events, feel emotionally and physically worn out.


*“We couldn’t help but take him there (hospital). What else we can do to get a little peace of mind! The stress, it keeps piling up. Day after day. We all get angry so often because of him…”*
(Subject I, a father taking care of a son with schizophrenia)

##### Helpless in Front of the Illness

Parents, who feel helpless in the face of the illness, are resigned to leave everything to fate.


*“The map of your fate is carved on the bones; you can never erase it, not even with a knife. We gave birth to him, and at that moment, the heavenly god painted his map of fate…His having this illness must have something to do with god’s will or karma, and that’s something you simply have to accept…”*
(Subject C, a mother taking care of a son with schizophrenia)

##### Desperate to Free Themselves from the Endless Worry and Exhaustion

Parents are desperate to free themselves from the shackles of endless worry and exhaustion.

*“When he lets himself go with those explosive tantrums, I hate him so much that I want to strangle him to death. I have told him more than once, ‘I don’t know when, but the time will come. Then it’s you or I, one of us has to die!’ It’s so unbearable seeing him like that. You know what I mean? Strangle him to death like this.”* (in Taiwanese) (Subject F, a mother taking care of a son with schizophrenia)

##### Hospitalization as a Temporary Relieving of Stress

Parents consider hospitalization is a temporary vent of their stress and pain.


*“I don’t mind spending some money taking her to the hospital. There are people taking good care of her, and that makes me feel more relieved. There she takes her medicine on time. At home, she makes no effort to take medicine, and she gets worse.”*
(Subject D, a mother taking care of a daughter with schizophrenia)

#### 3.2.3. Theme Three: Parents Consider Taking the Best Care of Their Children with Schizophrenia as Their Lifelong Responsibility

In addition to human nature and them being blood relations, parents are asked by society and under culture, to take the responsibility of taking care of their ill children. Two sub-themes were included:

##### Dedicate Themselves to Taking Care of Children

Mothers used to dedicate themselves to taking care of their children.


*“You feel sorry for him even when he is quite okay, having no condition. Regardless of who he is and how people feel about him, I am his mother. I never hate him so that I stop taking care of him. You know what? Being a mother, I am always the one taking care of his needs, before others notice him…He does have brothers and sisters, but they all have their own families to keep them busy. It is always and only me when he needs anything.”*
(Subject F, a mother taking care of a son with schizophrenia)

##### Prevent Ill Children from Becoming a Burden to Siblings

Parents endeavor to help their daughters develop skills for living independently. They do their best to avoid their sons becoming a long-term burden to their siblings.


*“She is now thirty. It would be great if she were able to marry someone who is willing to take care of her, but I know well that marrying her is bringing burdens to one’s family. After all, men, I mean most men, would not consider marrying a girl with an illness like her. So, it’s better to encourage her to be able to live by herself without depending on others…. I spend more time with her than others in my family. After all, her elder brother needs to work. So, I keep her company as best as I can…She doesn’t really have friends, and there is nothing much to talk about with her elder sister…Her elder sister is married and has her own family to take care of. We try our best to take good care of her; we’d be happy as long as she doesn’t catch a cold or become seriously ill. We also find time to take care of ourselves…It’s important to keep ourselves healthy as we are getting older.”*
(Subject A, a father taking care of a daughter with schizophrenia)

*“His elder brother fears of his wife so much…His wife would rather die than taking care my (ill) son. That’s just fine with me! His elder brother is a good person though, his wife is the problem, always nagging at her husband about this and that, lecturing and mocking him as if he is a child. I don’t need them to help me in taking care of my (ill) son.”* (in Taiwanese) (Subject E, a mother taking care of a son with schizophrenia)

#### 3.2.4. Theme Four: Parents Strive to Find Support That Brings Them Hope among Desperation

With the stigma and stereotypes imposed on people living with mental illness, parents develop a feeling of shame that is aggravated by their inability to figure out *“why my child?”* During the early stage of their children’s illness, they are therefore likely to seek answers and hope from folk medicine/treatment or religious healing. Religious belief particularly, can be a crucial source of strengthening parents to keep taking care of their ill children in spite of stress and pain. Three sub-themes were included:

##### Seek Hopes in Folk Medicine to Ease Helplessness

Parents seeks hopes in folk medicine or religious healing to ease their feeling of helplessness.


*“We pray to gods…spent quite a fortune …holding one rite after another…each costs over one hundred thousand dollars…We burned a lot of spirit paper money…Some of them, I know, are nothing but superstitions, but when you are in my situation, you do what people tell you to.”*
(Subject B, a father taking care of a son with schizophrenia)

##### Trust Modern Medical Treatment

Parents learn from experiences to trust modern medical treatment.


*“Her grandma keeps urging me, ‘Marry her to someone. That fixes everything.’ How is that possible? How could marriage help cure her illness?... People say that it’s called (pause)…“expelling bad luck with jubilance.” Wedding brings jubilance, and jubilance drives illness away. That’s just an old convention tale! Pure nonsense… For her illness, we go to doctors and take medicine. That’s what helps her control her illness, makes her stable. And once she is stable, she becomes a normal person.”*
(Subject A, a father taking care of a daughter with schizophrenia)

##### Seek Supports to Keep Taking Care of Children

Parents strive to seek supports and consolation to keep them taking care of their children.


*“This son of mine, he never ceases to cause me headache. Wherever I go, I can do worship and pray to gods. It’s all because of Buddha’s mercy that I am still here to take good care of my son. If I were not here, he would have died long time ago…Indeed, thanks to Buddha, I am able to save my son.”*
(Subject H, a mother taking care of a son with schizophrenia)

As the four main themes above indicate, there is no significant gender difference in the attitudes of parents toward their sons and daughters suffering from schizophrenia. While in terms of providing accommodation, assuming the responsibility of lifelong care, and seeking support, gender difference exerts no significant impact on the attitudes of parents living with adult children with schizophrenia. Nevertheless, issues related to family ethics or values, notably the aforementioned, needing to carry on the family name and the responsibility of continuing to take care of mentally ill family members, are marked with significant gender differences.

#### 3.2.5. Theme Five: Dictated by Traditional Chinese Cultural Values, Parents Continue to Expect Their Sons with Schizophrenia to Carry on the Family Name

Under the influence of Chinese cultural and family values, parents continue to keep the importance of their need to carry on the family name. As a result, adult ill sons are still expected to take this specific responsibility. Two sub-themes were included:

##### Carrying on the Family Name

Parents, under the influence of traditional family value, continue to believe their sons can build family through marriage, which is an essential link for carrying on the family name.


*“No matter what, we as parent still look forward to seeing our son reaching this crucial stage in life: marrying a wife and having their own children…It’s quite alright that he doesn’t get married. He has an elder brother, who can be married and having a son to avoid the family blood from broken. My grandson is now 23 or 24 years old. Mature for marriage. He is getting married soon!”*
(Subject E, a mother taking care of a son with schizophrenia)

##### Relief of Having Grandsons

Carrying on the family name is not an issue for mothers whose ill sons have already given them grandsons.


*“I don’t mind that he stays and dies in the hospital. We’ll take care of the cremation, burial, funeral and whatever is necessary after his death. His son has grown up now. We are lucky to have a grandson. Some don’t even have a chance to get married, and many of them do not get married nowadays”*
(Subject H, a mother taking care of a son with schizophrenia)

#### 3.2.6. Theme Six: Society as a Whole Expects Males to Be the “Head of the Family”

As prescribed by traditional Chinese culture, male family members are asked to fulfill the role of breadwinner as “the head of the family.” Illness of their children forces parents to adjust this expectation. Parental expectation, however, continues to vary in line with gender; sons should take more responsibility and have greater expectations. Two sub-themes were included:

##### Adjust Their Expectations of Their Ill Sons

Parents learn to re-examine and adjust their expectations of their ill sons.


*“I wouldn’t complain about being alone if he and his brother were doing well, but he’s like this now, nothing much we can expect from him. My greatest hope now is that he can learn to think straight, both for himself and for others. That’s it. I don’t expect him to make a fortune. Just be an ordinary person living an ordinary life. No need to be a great man. Be content with being ordinary. That’s one way to be happy, day in and day out. It’s a blessing, being able to be ordinary.”*
(Subject F, a mother taking care of a son with schizophrenia)

##### Work–Family Balance

Parents confronted the work–family balance influenced by their son’s illness.


*“The hospital warns us not to leave him alone since he has the illness, but I just can’t be there for him anytime. My son supports to work for family, but he sick. So, I need to work for the family; without my job, there will be no income. I have to go to work, and then I’ll be worrying about him all the time at work.”*
(Subject J, a mother taking care of a son with schizophrenia)

#### 3.2.7. Theme Seven: In Traditional Chinese Family Ethics, Male Family Members Are Supposed to Assume the Responsibility of Caring for Sick Siblings

*“Who will be taking care of him/her when I am gone?”* is a major worry of most parents living with their adult children with schizophrenia. According to Chinese family ethics, healthy brothers are expected to “inherit” (take over) their parents’ responsibility to take care of their single siblings. Moreover, based on the emphasis of Chinese culture on male superiority over women and seniority in the family, the eldest son is morally required to assume the responsibility, from which married daughters are usually exempted. Two sub-themes were included.

##### Healthy Sons to Take Care of the Ill Sibling

Parents believes that it is the responsibility of their healthy sons to take care of the ill siblings.


*“I’ve lucky to have son. I don’t care, I mean how could I? About what will happen. I’m getting old and then I die. It’s a good thing that you see and know nothing after you die. After I’m gone, his younger brother, a normal person, is supposed to take care of him. If that does not happen, the government will look after him…”*
(Subject B, a father taking care of a son with schizophrenia)

##### The Responsibility of Taking Care of Ill Children for Life

Parents require family members to assume the responsibility of taking care of their ill children for life.


*“If something terrible happened to her, we would handle her funeral. If we die before she does, then it’s up to her, but she no longer has her elder brother. There is only her elder sister left to deal with her (suggesting that the elder brother would have had to be the primary caregiver of his younger sister if he had not died)*
(Subject D, a mother taking care of a daughter with schizophrenia)

## 4. Discussion

This study indicated that, regardless of gender, adult sons and daughters with schizophrenia remain irreplaceable to their fathers and mothers, playing the role of beloved children for their lifetime. Gender difference exerts no influence on parents’ attitude toward their ill children, which is similar to other studies [29,30]. Mental illness does not affect the parent–child bond as reflected in these words: *“He/she fills my life with meaning and purpose”* and *“Although I keep growing old and weak; he/she is always an integral part of me”* [31]. While the previous study focused on the aging caregivers of middle age and older family members with intellectual disability; in this study, focused on parents taking care of adult children with schizophrenia, the common theme observed is that of the inseparable bond between mentally ill children and their parents.

Gender apparently makes a difference in parents’ attitudes toward their sons and daughters, but only in the issue of carrying on the family name in Chinese-speaking society. Based on traditional Chinese cultural and family values, carrying on one’s family name remains a crucial concern that undergoes minor change over time [32]. In Tseng and colleagues’ [33] study adopting a phenomenological approach to examine the influence of gender difference on the aging experiences of the elderly in Taiwan, they suggested that older Taiwanese women tend to consider carrying on the family name as a personal mission in life. They were therefore particularly worried about their children’s marriage and having grandchildren. This finding echoes the fifth theme in this study that parents continue to expect their sons with schizophrenia to fulfill the responsibility of carrying on the family name. Parents in this study, however, show no such expectation for their ill daughters. Similar findings can be found in the study by Peng et al., [5] that the caring burden may relate to kinship types of patients with schizophrenia in Chinese society. Individuals with mental illness usually have lower marriage rate, but this phenomena happened in male more than female patients due to males carrying a heavier responsibility for providing a family living, while females are asked to meet housekeeping tasks [34]. Therefore, both in Chinese society and other countries in the world, marriage for mental illness patients is desired but difficult.

In theme one, gender difference showed no significant influence on parents’ need to re-examine the roles and wishes that their ill children may not be able to accomplish. Schizophrenia is a mental illness that emerges typically in adolescence and early adulthood. In addition to cognitive deficits and impairing daily functioning, schizophrenia is a chronic condition with attached stigma causing patients to fail to build interpersonal relationships, play various social roles, and stopping their dreams. The children of patients with schizophrenia have a lower fertility rate than the general population [35,36]. There is a difficulty or even inability of their children with schizophrenia to fulfill expected roles, and it is hard for them to build families through marriage; parents have to learn to reduce or re-adjust their expectations of their sons and daughters. Also, caregivers’ burdens may vary in different countries due to different health care systems [37].

While parents are willing to adjust their expectations of their sons and daughters, gender becomes an issue when the expectation is related to taking responsibility for the family. Chinese culture emphasizes the notion that “men are breadwinners and women are homemakers.” Men are expected to achieve success and win recognition to make their families proud [8,38]. Women, on the other hand, are asked to take pride in supporting husbands and parenting children; quite a few studies on foreign brides indicate that immigrant wives are even required to take care of all the male family members [39,40]. It is therefore understandable that parents still expected their ill sons to assume the role of “the head of the family.”

As revealed in themes two, three, and four, to parents, taking ill children to hospital offers a temporary break from accumulated stress and exhaustion; taking care of their ill children is a lifelong responsibility, and external support is necessary to ease their desperation. These three themes remained unchanged by gender difference. Nevertheless, when the issue is about the responsibility of taking care of ill siblings, the gender difference issue showed again in theme eight. Aggressive or violent behavioral problems during acute periods make caring for patients with schizophrenia a daunting task for family members [41], who may feel so overwhelmed that they would rather choose to place their loved ones in hospitals. With the rising trend of deinstitutionalization, treatment of patients with mental illnesses has become community oriented. Studies, however, continue to find family members, especially parents, playing the role of primary caregiver as they recognize the role as their familial obligation [42,43,44].

Family members tend to attribute the causes of illness to life experience, culture or religion [6]; they also attempt to find support and spiritual consolation in religious belief or folk treatments. These findings reported in previous literature are consistent with the results in this study. However, in Taiwanese society, family is cemented by traditional cultural values, which require each member conscientiously to fulfil his or her prescribed roles by gender. Males members are accordingly expected [23] to assume greater responsibility as heads of family [38,45] including carrying on the family name and taking care of ill siblings. From a different point of view, in a comprehensive big-scale study with long-term follow up in Finland, married schizophrenia patients had a better quality of life related to partial help in life from their spouses. The same study indicated that female patients showed better adaption in society, especially in the married group; in contrast, single male patients were more likely to live apart from the challenges of society [46].

The study adopted purpose sampling to enroll subjects from a medical center in central Taiwan with a limited sample size. Discrepancies may exist between the enrolled subjects because of their various living regions, religious beliefs and susceptibility to traditional concepts and values, those factors may influence their knowledge and interpretation of the illness. Findings of this study may lack generalizability due to these limitations. Moreover, patients with schizophrenia in the study had considerable differences in sex, age-at-onset and history of the illness. Therefore, future studies should consider expanding the sample size and minimizing differences in age-at-onset and history of the illness among patients.

## 5. Conclusions

Exploring the influences of gender difference on the attitudes of parents taking care of adult children with schizophrenia, the study finds that they are unable to go against traditional cultural and family values; parents still expected their ill sons to be able to fulfill the responsibility of carrying on the family name. Male family members were expected to take the role as the head of the family especially, and to take over their parents’ responsibility of taking care of their ill siblings. Therefore, it is not surprising that some parents choose to let their ill sons marry immigrant wives whose major functions include bearing grandchildren to carry on the family name and serving as lifelong caregivers of their husbands. In a society which places great emphasis on family ethics and prescribed gender roles, women are expected to conform to the stereotype of being the caregivers for their families. These findings can be used to help clinical professionals understand the difficulties and concerns of parents taking care of their adult children with schizophrenia with greater empathy and to provide them more responsive assistance.

## Figures and Tables

**Figure 1 healthcare-09-00836-f001:**
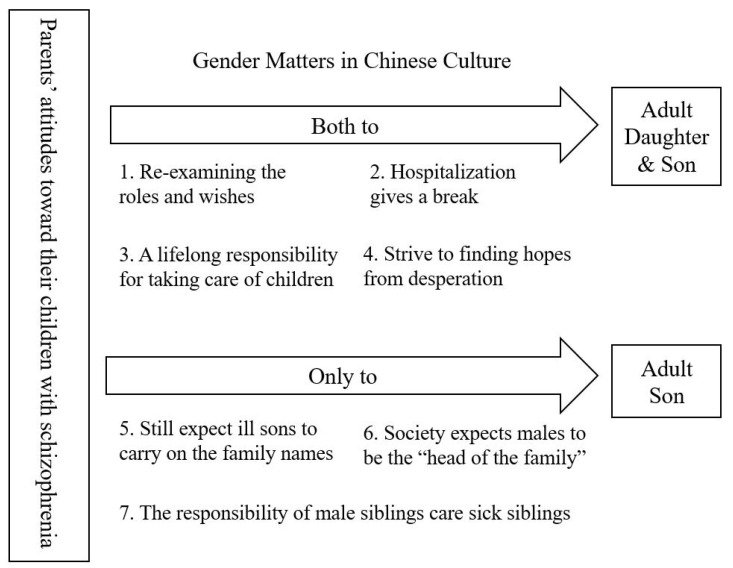
Gender Differences on the Parents’ Attitudes Toward Adult Children with Schizophrenia.

**Table 1 healthcare-09-00836-t001:** Interview Guide.

No.	Suggested Topic/Question
1	Would you like to talk about the process of your child’s development of the illness?
2	What are the impacts of your child’s illness on your family? How do you cope with those impacts?
3	How would you describe your daily life living conditions? Are there any additional impacts?
4	What concerned/worried you the most after your child became ill?
5	What are your plans to face an illness that needs long-term care?
6	Does your child’s illness change your expectation(s) of him or her?
7	What were your expectations before his/her illness? How about after the illness?
8	Before the illness, was your child able to fulfill his or her responsibilities and roles at school, at work, and/or in family. This, however, became something he or she is unable to accomplish or to do well. How do you feel about this change?

**Table 2 healthcare-09-00836-t002:** Essential Demographic Information of Research Subjects.

Cases	Parent as Primary Caregiver				Adult Child with Schizophrenia				
Code	Role	Age	Education	Job	Role	Age	Education	Marital Status	History of Schizophrenia by Year
A	Father	60	High School	Retired	Daughter	29	Junior High	Single	15
B	Father	65	Primary	Retired	Son	40	Junior High	Single	20
C	Mother	53	Vocational School	Hairdresser	Son	25	High School	Single	7
D	Mother	60	Junior High	Homemaker	Daughter	39	Junior High	Divorced	15
E	Mother	79	Illiterate	Homemaker	Son	48	Junior College	Single	25
F	Mother	66	Junior High	Farmer	Son	46	Vocational School	Divorced	15
G	Mother	73	Junior High	Homemaker	Son	51	Junior High	Single	25
H	Mother	90	Illiterate	Homemaker	Son	54	Junior High	Divorced	25
I	Father	54	Junior High	Laborer	Son	30	College	Single	3
J	Mother	62	Junior High	Temp Worker	Son	39	Primary	Divorced	20

**Table 3 healthcare-09-00836-t003:** Main Themes and Sub-Themes Identified by Data Analysis.

Main Theme	Sub-Theme
1. Parents do recognize the need to re-examine the roles and wishes their children are unlikely to be able to fulfill	(1) Experience shame and learn to accept
	(2) Forced to give up expectations
	(3) Fail to be independent on their own
2. Hospitalization gives parents a break after long-term mental and physical burnout	(1) Parents feel emotionally and physically worn out
	(2) Helpless in the face of the illness
	(3) Desperate to free themselves from the endless worry and exhaustion
	(4) Hospitalization as a temporary exit out of stress
3. Parents consider taking the best care of their children with schizophrenia as their lifelong responsibility	(1) Dedicate themselves to taking care of children
	(2) Prevent sons from becoming a burden to siblings
4. Parents strive to find support that brings hope among desperation	(1) Seek hope in folk medicine to ease helplessness
	(2) Trust modern medical treatment
	(3) Seek support to keep taking care of children
5. Dictated by traditional Chinese cultural values, parents continue to expect their ill sons to carry on the family names	(1) Carrying on the family name
	(2) Relief from having grandson
6. Society as a whole expects males to be the “head of the family”	(1) Adjust expectations of ill sons
	(2) Work–family balance
7. In traditional Chinese family ethics, male family members are supposed to assume the responsibility of caring for sick siblings	(1) Healthy sons to take care of the ill sibling
	(2) The responsibility of taking care of ill children for life

## Data Availability

These study data are deidentified participant data. The data that support the findings of this study are available beginning 12 months and ending 36 months following the article publication from the corresponding author, W-FM, upon reasonable request at lhdaisy@mail.cmu.edu.tw.

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
