# Peer review of "Gender Differences in the Attitudes of Parents Living with Adult Children with Schizophrenia"

_healthcare, 2021, doi:10.3390/healthcare9070836_

Round 1
Reviewer 1 Report
This is "The Gender Difference on the Attitudes of Parents Living with Adult Children with Schizophrenia." I think this is a meaningful and important study in the field of nursing. However, I hope that the extracted themes will be arranged in pictures for the readers. Also, it would be helpful for researchers in other countries to write in the introduction and discussion section comparing the situation of parents living with adult children with schizophrenia in Chinese and parents living with adult children with schizophrenia in other countries in detail.
Author Response
Response Letter (Manuscript ID: healthcare-1264228)
To Reviewer
Thanks for reviewer’s comments. We are very grateful to you for all the insightful suggestions that help enhance the quality of our paper. Our responses to your suggestions and questions are outlined below, with reference to the appropriate pages in the text. Revisions in the text and tables have been highlighted in RED. Thank you.
Comments and Suggestions for Authors
- This is "The Gender Difference on the Attitudes of Parents Living with Adult Children with Schizophrenia." I think this is a meaningful and important study in the field of nursing. However, I hope that the extracted themes will be arranged in pictures for the readers.
Response:
Thanks for reviewer’s suggestion. The authors agree and design a new figure 1 to clarify the themes whose themes with and without gender differences. Thanks for reviewer’s this comment. We believe Figure I makes the themes clearer. Please see the Figure 1 (page 6, line 170-171) in the manuscript.
- Also, it would be helpful for researchers in other countries to write in the introduction and discussion section comparing the situation of parents living with adult children with schizophrenia in Chinese and parents living with adult children with schizophrenia in other countries in detail.
Response:
Thanks the reviewer very much for your suggestion. The authors added 7 new references and revised text regarding parents living with adult children with schizophrenia in other countries and cited those in the introduction and discussion section. We believe these citations make this manuscript completely. Please see the Introduction: page 2 (line 68-69; 77-80; 86-88) and Discussion: page 10 (line 397-400), page 11 (line 401-402), page 11 (line 412-413) in the paper. Thanks for reviewer’s comment again.

Reviewer 2 Report
This manuscript presents gender based qualitative differences in attitudes of parents of adult children living with major mental illness.
Please articulte a clear puroose and a set of reesarch questions that this manuscript will answer right at the end of th eintro. These need to be derived from the lit review
Provide citations for the well written qualitative methdos and choices,, from the liteatrur Might want to try OBSSR documents on qualitative methods (available online at OBSSR
Consider putting Table 3 in the supplemental materials section.
Not clear where the four themse wiht no gender differents are located. Please clarify.
Author Response
Response Letter (Manuscript ID: healthcare-1264228)
To Reviewer
Thank you very much for taking time from your busy schedule to comment on our manuscript. Revisions in the text have also been highlighted in RED. Thank you.
Comments and Suggestions for Authors
1.This manuscript presents gender based qualitative differences in attitudes of parents of adult children living with major mental illness. Please articulate a clear purpose and a set of research questions that this manuscript will answer right at the end of the entire. These need to be derived from the lit review.
Response:
Thanks the reviewer very much for your positive remarks. The authors agree that the important of issues related to attitude on gender. The research purpose, research questions, and literature background are all revised that making whole manuscript consistency. The changes we marked on red words. Thanks reviewer for taking time to review this paper.
2.Provide citations for the well written qualitative methods and choices, from the literature Might want to try OBSSR documents on qualitative methods (available online at OBSSR).
Response:
Thanks for reviewer’s suggestions. The authors agree and added new two citations in the methods including one from the online document at OBSSR. We believe these citations make this method completely. Please see the page 2-3 of paper.
3.Consider putting Table 3 in the supplemental materials section.
Response:
Thanks for reviewer’s this suggestion. Since this paper will be published online as open access if the paper is accepted by the Journal. Online space may not a limited issue. The authors keep the table 3 in the manuscript right now. However, the authors will agree to put the table 3 in supplemental materials section if the Journal suggests no matter the reasons in future.
- Not clear where the four themes with no gender differences are located. Please clarify.
Response:
Thanks for reviewer’s comment. The authors design a new figure 1 to clarify the themes whose themes with and without gender differences. Thanks for reviewer’s this comment. Please see the Figure 1 (page 6) in the manuscript.
